# LABs Fermentation Side-Product Positively Influences Rhizosphere and Plant Growth in Greenhouse Lettuce and Tomatoes

Gabriele Bellotti [1], Eren Taskin [1], Simone Sello [2], Cristina Sudiro [2], Rossella Bortolaso [2], Francesca Bandini [1], Maria Chiara Guerrieri [1], Pier Sandro Cocconcelli [1], Francesco Vuolo [3,*] and Edoardo Puglisi [1]

[1] Department for Sustainable Food Process (DiSTAS), Faculty of Agriculture, Food and Environmental Sciences, Università Cattolica del Sacro Cuore, 29121 Piacenza, PC, Italy
[2] Landlab S.r.l., 36050 Quinto Vicentino, VI, Italy
[3] Sacco S.r.l., Via Alessandro Manzoni 29/A, 22071 Cadorago, CO, Italy
* Correspondence: f.vuolo@saccosrl.it

**Abstract:** New agronomical policies aim to achieve greener agricultural systems, sustainable fertilizers and fungicides, a reduction in Greenhouse gases (GHG), and an increase in circular economic models. In this context, new solutions are needed for the market, but it is necessary to carefully assess both their efficacy and their ecological impact. Previously, we reported the biostimulatory activity on soil microbiome for a side-product from Lactic Acid Bacteria (LABs) fermentation: a concentrated post-centrifugation eluate. In the present study, we investigated whether this solution could partially substitute mineral N (N70% + N30% from eluate) in a fertigation (N100% vs. N70%) regime for tomato and lettuce under greenhouse conditions. The impact of the application was investigated through plant physiological parameters (number and weight of ripened fruits, shoots, and roots biomass) and biodiversity of the rhizosphere microbial composition of bacteria and fungi (High-Throughput Sequencing—HTS). The eluate (i) enhanced the plant canopy in lettuce; (ii) increased the shoot/root biomass ratio in both tomato and lettuce; and (iii) increased the harvest and delayed fruit ripening in tomato. Moreover, we found a strong correlation between the eluate and the enrichment for OTUs of plant-growth-promoting microbes (PGPMs) such as *Sphingomonas sediminicola*, *Knoellia subterranean*, and *Funneliformis mosseae*. These findings suggest that integrating the eluate was beneficial for the plant growth, performance, and yield in both tomato and lettuce, and additionally, it enriched specialized functional microbial communities in the rhizosphere. Further studies will investigate the underlying mechanisms regulating the selective activity of the eluate toward PGPMs and its biostimulatory activity towards target crops.

**Keywords:** biofertilizer; biostimulant; plant science; agroecology; food production; circular economy; high-throughput sequencing; soil microbiome



## 1. Introduction

Conventional agriculture is heavily dependent on synthetic fertilizers, and in past years, to maximize crop yields, their use has been too vast, leading to reduction in Nutrient Use Efficiency (NUE), leaching, aquifer contamination, surface water eutrophication, increasing soil salinity, soil heavy metal contamination, ammonia volatilization, and emissions of GHG $N_2O$ [1–4].

In order to mitigate these effects and reverse this trend, governments have promoted new agricultural practices worldwide, aiming for substantial reductions in synthetic fertilizer usage and more eco-sustainable fertilization campaigns. In this way, new solutions (such as biofertilizers, biopesticides, and biostimulants) are highly requested to sustain crop yield and to balance the soil eco-system [5]. Many studies have already reported the successful use of these new classes of modern fertilizers to increase the soil organic

carbon (SOC) [6], to replace mineral N, to reduce $N_2O$ emissions [7], to be co-inoculants for beneficial microorganisms, and to biologically nourish and protect plants [8].

The pivotal value of these new fertilizers is their sustainability, which characterizes their entire value-chain. In fact, biofertilizers support food and feed production through a reduced application dose, which has an eco-sustainable influence on biogeochemical cycles, as well as a more rational delivery of plant nutrients [9,10]. Last, but not least, their lower requested volumes are linked to smaller carbon footprints for both their production and transportation.

Following this rationale, circular economy models are the ideal base to create new values for side-products, which could be the ingredients of new agro-biological solutions [1]. These models could offer a double solution to both the mitigation of climatic crisis as well as the supply of new raw materials and products in a troubled geopolitical scenario [11], complying perfectly with both the UE Green Deal policy as well as the Sustainable Development Goals (SDGs) of the United Nations (https://sdgs.un.org/goals accessed on 11 July 2022).

Although these important goals target the overall climate balance, the core of a healthy agricultural system relies on healthy soil with good physicochemical properties and also positive microbial communities, which are key to linking plant health and organic fertilizer effectiveness [12]. For these reasons, plant biologists and agronomists are also focusing their targets around PGPM soil compositions and influence in plant fitness [13,14] to study how the next biofertilizers will impact agro-ecosystems.

The present study shows how the LABs eluate could serve as an integration of mineral N fertilization by testing its effect on the harvest of tomato and lettuce plants. We chose it because in order to generate dairy/probiotics starter cultures, the fermentation industries produce a vast amount of this side-product, which actually retains several nutrients and represents a noble leftover that could be reintroduced into new economies, especially because it is produced with the highest food- and pharma-grade standards.

We show that, by replacing 30% of the total N requirement in these crops' fertilization practices, we reach comparable yields in lettuce and higher ones in tomato. Moreover, we report that the addition of the LABs eluate enriches plant rhizospheric soil with PGPMs. The latter results are in accordance with our previous study on the enrichment of PGPMs within bulk soil mediated by the LABs eluate [15]. Collectively, these data support the potential use of the eluate as an eco-sustainable, circular, and biostimulatory integration into conventional fertilization practices.

## 2. Materials and Methods

### 2.1. Eluate Preparation and Composition

The product used in this study was provided by Sacco S.r.l. and is a microbial eluate that is the end-of-fermentation broth retrieved from Lactic Acid Bacteria (LAB) production. It was harvested and concentrated under vacuum through heating at 95 °C, so that around 90% of the water is recovered. The full procedure is described and protected by the patent No. PCT/EP2021/080974. The eluate contains: N-P-K (4.5-1-0.5 % $w/w$), a concentration of lactic acid >15%, and micronutrients in traces. The eluate amino acidic composition is indicated in Table 1.

### 2.2. Experimental Design and Treatments

Pot experiments were conducted in a completely randomized design in greenhouse conditions using lettuce (*Lactuca sativa* L.-cv Gentile) and bushy tomato (*Solanum lycopersicum* L.-cv. sweet 'n' neat yellow) at LandLab S.r.l. Quinto Vicentino Vicenza, Italy. We selected the tomato as it is one of the most important industrial plants in Italy that is widely used for various purposes, and its environmental footprint may be reduced using recycled byproducts, thus rendering its production more efficient and sustainable. Lettuce was selected because it is commonly consumed worldwide as fresh greens and due to its recent use in vertical farming and hydroponics applications. These two plants are also

widely studied in the literature for such experiments. Therefore, selecting them allowed us to compare the use of the eluate to other conventional nutrition substitutes. Young lettuce plants were purchased at a local nursery, while tomato seeds were germinated in a commercial peat-based germination mix in a controlled environment until seedling stage. After 21 days, seedlings were randomly selected and transplanted in 3 L pots filled with a dedicated substrate (70% sand +30% soil). Lettuce and tomato plants were grown for 50 and 90 days, respectively, from mid-March, with a photoperiod of approximately 12–15 h with natural daylight and with day/night temperatures that spanned from 40 °C to 15 °C (Figure S1). The solar radiation was recorded using a weather station (Davis Vantage Pro2, Davis Instruments, Hayward, CA, USA) located at Landlab S.r.l., Quinto Vicentino, and the greenhouse air temperature was recorded with a datalogger (Keytag Kt1Mu, Askey, Leiderdorp, The Netherlands).

**Table 1.** Eluate amino acidic composition determined with HPLC analysis.

| Amino Acid | Content % |
|:---:|:---:|
| Alanine | 0.96 |
| Arginine | 0.40 |
| Asparagine (incl. Aspartic acid) | 1.07 |
| Hydroxyproline | 0.25 |
| Cysteine | 0.24 |
| Glutamic acid | 1.83 |
| Glycine | 0.55 |
| Histidine | 0.20 |
| Isoleucine | 0.46 |
| Leucine | 0.74 |
| Lysine | 0.77 |
| Methionine | 0.13 |
| Phenylalanine | 0.41 |
| Proline | 0.45 |
| Serine | 0.46 |
| Threonine | 0.41 |
| Tryptophan | 0.08 |
| Tyrosine | 0.29 |
| Valine | 0.62 |

The nutrition plans adopted for tomato and lettuce were the optimal model developed by Landlab S.r.l. in pot trials, matured over the course of years of research, and were considered as follows: lettuce $N-P_2O_5-K_2O$ 32.5-12.2-48.8 considering 80,000 plants/ha, and tomato $N-P_2O_5-K_2O$ 101.3-50.6-210.9 considering 50,000 plants/ha. Nutrients were provided via fertigation considering 2 events per week to supply plants 100% or 70% N. In detail, soluble fertilizers Novatec Solub 21 (Compo Expert), Hakaphos Basis 5 (Compo Expert), and Multi-K (Haifa) were balanced to provide the requested NPK, and they were dissolved in a water volume to provide 100 mL of solution per plant per fertigation. The solution was provided by hand, plant by plant. The eluate was considered to substitute 30% of the mineral N and reach the condition of 100% N. The resulting entries were the following: N100%, N70%, N70% + N30% with eluate. For every condition, 4 replicates made in 4 pots each were considered.

Irrigation was provided with a water volume that followed plant request, depending on the weather and plant growth stage.

### 2.3. Agronomic Parameters

The growth of lettuce plants was monitored by capturing high-definition pictures of all plants 3 times during the trial: on the day of first eluate application, 7 days after transplant (T0); 23 days after T0 (T1); and 44 days after T0 (T2). The images were processed and analyzed as described in [16]. In brief, pictures were acquired with a reflex camera (Canon 2000D) mounted on a box that allowed pictures to be taken from above plants, at a

fixed distance and illumination for all dates. Images were then analyzed using WinCAM (Regents Instruments Inc., Sainte Foy, Quebec, Canada) to generate values in cm$^2$ (sqcm) to quantify the canopy of plants.

Tomato production was assessed by harvesting fruits 77 and 90 days after trans-plant. Ripe fruits were counted and weighed on the two dates, and unripe tomatoes were assessed on the second date to estimate the total potential fruit production.

Shoot and root fresh biomass were assessed at the end of the trial by separately weighing the two parts of every plant.

*2.4. Rhizosphere DNA Extraction and Amplification*

Roots were removed from the pots at the end of the trial, 50 and 90 days after transplanting for lettuce and tomato, respectively. The bulk soil was removed by manual shaking, and the soil in the vicinity of roots was collected. Subsamples from the 4 pots in every replicate were pooled together. The total DNA was extracted with the FastDNA™ SPIN Kit for Soil (MP Biomedicals, Santa Ana CA, USA) according to the manufacturer's protocol from 500 mg of rhizosphere soil. Isolated DNA in each sample was quantified using the Quant-iT™ HS ds-DNA assay kit (Invitrogen, Waltham, MA, USA) with a QuBit™ fluorometer. Subsequently, DNA quality was verified on 1% agarose gel before PCR amplification.

The rhizosphere DNA was amplified via PCR to assess the microbial composition of each sample (V3–V4 hypervariable regions of the 16S rRNA gene for bacteria, Internal Transcribed Spacer 1 (ITS1) region of ribosomal DNA (rDNA) for fungi). The following sets of primers were used: 343f (5′-TACGGRAGGCAGCAG-3′) and 802r (5′-TACNVGGGTWTCTAATCC-3′) and ITS-1 (5′-TCCGTAGGTGAACCTGCGG-3′) and ITS-2 (5′-GCTGCGTTCTTCATCGATGC-3′), for bacteria and fungi, respectively. Reaction conditions, concentrations, and thermocycling profiles were previously detailed in [15,17], and they were followed without further modifications. Two-step PCRs were performed independently for each sample as reported in [18,19]. Briefly, the first one was carried for 20 cycles and the second one for 10 cycles, but the PCR mix included forward barcoded primers with a unique nucleotide code for the identification of each sample after the sequencing [20]. After the two PCRs, amplicons obtained were quantified as previously described, and the bacterial and fungal amplicons were then combined in two different pools. Each sample was pooled in equimolar ratio (30 ng of amplicons). The two pools were then purified with the solid phase reversible immobilizations (SPRI) method using Agencourt AMPure XP kit (Beckman Coulter, Milano, Italy) and shipped to Fasteris S. A. (Geneva, Switzerland) for sequencing using MiSeq Illumina technology (Illumina Inc, San Diego, CA, USA). Amplicon library preparation was done with the TruSeq DNA sample preparation kit (Illumina Inc, San Diego, CA, USA).

*2.5. Amplicon Sequences Data Preparation*

Sequences of 300 bp paired-end reads were generated, setting a minimum overlap of 30 bp between read pairs and a maximum of 2 mismatches allowed. The obtained sequences were demultiplexed according to the indexes and primers codes; the software used was the Fastx-toolkit (http://hannonlab.cshl.edu/fastx_toolkit/, accessed on 1 July 2022). Raw reads were aligned, and the amplicon sequences were generated by the PANDAseq software [21]. Chimeric sequences such as homopolymers > 10 bp and sequences that did not align with the target one (V3–V4 for bacteria and ITS1 for fungi) were discarded. Those sequences were removed using Mothur version 1.32.0 and the UCHIME algorithm with the UNITE database v6, respectively, for bacteria and fungi. The high-quality sequences obtained were processed with two different approaches, the operational taxonomic unit (OTU) and the taxonomy-based approach. For V3–V4 regions, OTUs and taxonomy matrixes were analyzed using Mothur v.1.32.1 [22], while statistical analyses were done by R version 3.0.0 (http://www.R-project.org/, accessed on 1 July 2022) supplemented with the Vegan package [23]. OTUs belonging to ITS1 amplicons were determined in UPARSE [24],

as no aligned databases are available for ITS1. Mothur was then set with a minimum length of 120 bp and no upper length limit due to ITS variability.

### 2.6. Statistical Analysis

The statistical analysis (software XLSTAT by Addinsoft) for agronomic traits was conducted by considering the average value of every assessed parameter for every plot. The statistical analysis was performed for every parameter by means of one-way analysis of variance (one-way ANOVA) with Duncan Test ($\alpha$) = 0.05. For every trait, at least one letter in common indicates no significant difference according to the Duncan test.

HTS sequences obtained were statistically analyzed in Mothur and R integrated with the Vegan package following the operational taxonomic unit (OTU) and taxonomy-based approach. The two approaches included the analysis of the microbial $\alpha$-diversity based on the Shannon's Index, the Observed Richness (S), the Simpson's Diversity Index (D), and the Chao's Index. The average linkage algorithm was applied at different taxonomic levels to visualize the hierarchical clustering of the sequences. Furthermore, the unconstrained sample grouping was assessed by Principal Component Analysis (PCA), while the Canonical Correspondence Analysis (CCA) was used to visualize the significance of different treatments on the analyzed diversity (constrained variance). Moreover, to examinate the differential abundances accounting for at least 1% of reads for a given sample, the Metastats script coupled with the FDR test for means comparison [25,26] was used to identify which OTUs were significantly different between different treatments. Finally, the OTU sequences found to be significantly different were submitted to the RDP (Ribosomal Database Project) and NCBI (National Center for Biotechnology Information) databases for bacterial OTUs, while Mycobank and NCBI databases were used for fungal OTUs.

## 3. Results

### 3.1. Effect of the Eluate on Plant Growth and Production

The use of the eluate aimed to reduce the application of mineral nitrogen by taking advantage of the N naturally present in the byproduct to limit the exploitation of the price-increasing nutrient. Following the nutrition plan for lettuce and tomato, respectively, 0.22 and 0.54 g eluate per plant per application were supplied, thus, 232.7 kg/ha and 724.9 kg/ha. The effect of the use of the eluate as a N source on plant growth and production was assessed considering the growth of lettuce canopy, the fruit production for tomato, and shoot and root biomass for both plant species.

#### 3.1.1. Enhancement of Lettuce Growth

Lettuce growth was followed considering its canopy at three timepoints: the day of the first eluate application (T0) and 23 (T1) and 44 (T2) days later. The three conditions recorded very similar values at T0 (Table S2), but the differences between T0–T1, T1–T2, and in particular T0–T2, indicate that the eluate had a positive effect on plant growth (Figure 1; Table S3). In fact, N70% + N30% with eluate gave in the three timespans, respectively, +9.7%, −4.2%, and +3.2% versus N100%, with an overall improvement of plant canopy. N70% gave lower values than the other two conditions, as expected, and the total growth was significantly impaired compared to N70% + N30% with eluate (−12.9%).

The biomass of lettuce plants and the respective roots were also assessed. Results indicate that N70% + N30% with eluate gave +6.0% increase in terms of shoot biomass compared to N100%, while the root biomass was −2.7% versus the control. On the other hand, N70% gave −7.9% shoot biomass and +5.9% root biomass (although the difference was not significant in the latter case), suggesting that this condition induced plants to produce more roots to search for nutrients in the soil (Table 2).

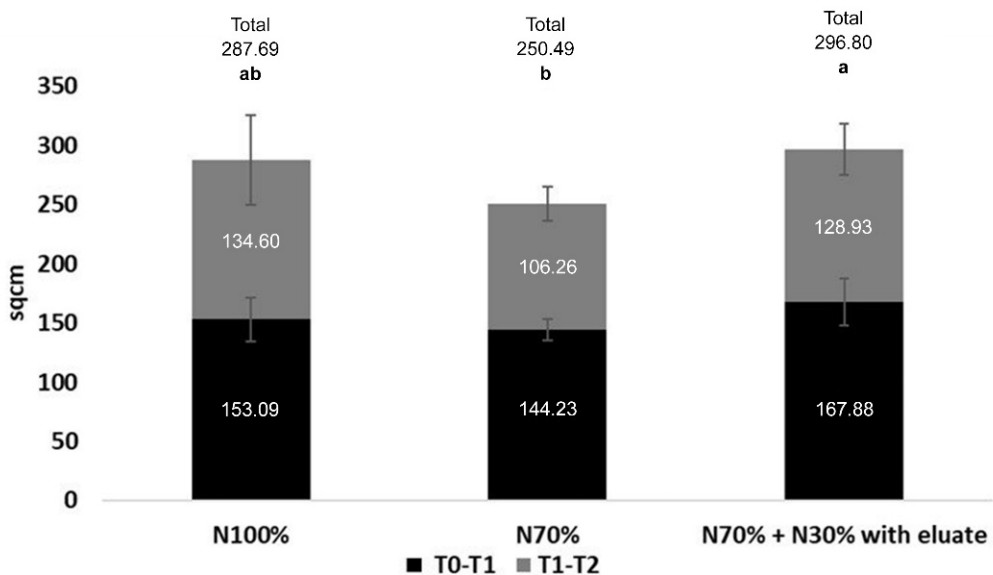

**Figure 1.** Effect of the eluate on plant growth. The replacement of 30%N with the eluate filled the gap and surpassed N100% in the first growth phase (dark grey) and in total growth. The numbers in the bars are expressed in sqcm and indicate the difference between T0–T1 and T1–T2, while the total growth is shown above the bars. Thin light grey bars indicate 95% confidence interval. Bars labeled with different letters differ significantly ($p < 0.05$, Duncan test).

**Table 2.** Lettuce shoots and roots fresh biomass. The addition of the eluate to the fertigation plan improved the weight of lettuce plants, while it reduced the weight of roots. A reduced N supply reduced the shoot biomass and increased root weight. Values ±95% confidence interval are expressed in g. Different letters differ significantly ($p < 0.05$, Duncan test).

| Fertilization Level | Shoot Fresh Biomass (g) | Root Fresh Biomass (g) |
|---|---|---|
| N100% | 173.57 (±14.23) ab | 17.96 (±2.31) |
| N70% | 159.82 (±3.75) b | 19.01 (±1.93) |
| N70% + N30% from eluate | 183.99 (±13.77) a | 17.47 (±1.72) |

3.1.2. Enhancement of Tomato Growth and Production

The same approach of reducing the use of mineral N was also considered for tomato, where the performance of the eluate in terms of fruit production and plant growth was considered. Ripe fruits were harvested, counted, and weighed twice, 77 and 90 days after transplant, and unripe tomatoes were assessed at the end of the trial. Results indicate that the reduction in available N (N70%) gave the highest number of ripe fruits per plant at the first harvest, with a subsequent higher yield, indicating an anticipated fruit maturation (Figure 2; Table S3). In fact, N70% gave +15.7% and +19.4% versus N100% in terms of fruit number and weight, respectively. Conversely, N70% + N30% with eluate resulted in a delayed ripening, with −11.3% in number and −14.6% in weight compared to N100%. At the second harvest, the use of the eluate increased both parameters, with +7.2% fruits and +9.9% weight compared to N100%. Considering unripe fruits in the total amount of produced fruits, N70% + N30% with eluate recorded the highest production, with +18.5% fruits and +11.6% weight compared to N100% and +19.7% and +15.7% compared to N70%.

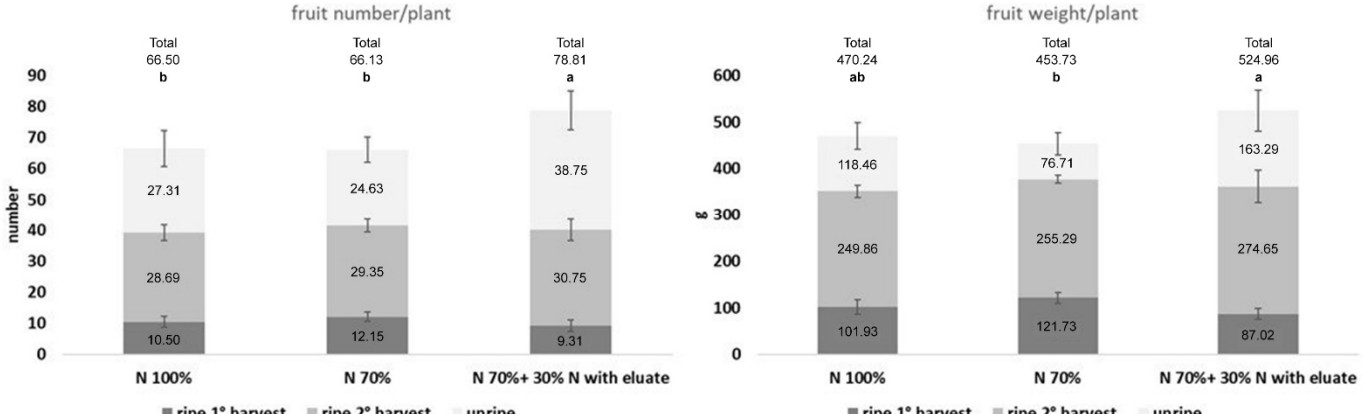

**Figure 2.** Effect of the eluate on tomato production. Ripe fruits (dark grey and mid grey) were harvested twice, while unripe fruits (light grey) were harvested at the end of the trial, at the second harvest. The reduction in N (N70%) caused an anticipation of fruit ripening (**left** box, 1° harvest) and an increase in the yield (**right** box, 1° harvest). Conversely, the addition of the eluate to reach full nutrition delayed both parameters, but it caused a significant increase in the total fruit number and a high rise in the total yield, also compared with N100%. Thin black bars indicate the 95% confidence interval. The total production is shown above the bars. Bars labeled with different letters differ significantly ($p < 0.05$, Duncan test).

The weight of plant shoots and roots was evaluated at the end of the trial by separating and weighing them. Results indicate that the addition of the eluate to the nutrition plan gave +12.0% and −7.9% shoot and root biomass, respectively, while both parameters for N70% were reduced (−7.9% shoot and −23.2% root weight) compared to N100% (Table 3).

**Table 3.** Tomato shoots and roots fresh biomass. The addition of the eluate to fill the N gap with N100% gave the highest value in terms of shoot biomass and reduced the growth of roots compared to full nutrition. Values 95% confidence interval are expressed in g. Different letters differ significantly ($p < 0.05$, Duncan test).

| Fertilization Level | Shoot Fresh Biomass (g) | Root Fresh Biomass (g) |
|---|---|---|
| N100% | 107.69 (±7.26) | 22.53 (±4.31) |
| N70% | 99.24 (±8.51) | 17.30 (±2.62) |
| N70% + N30% from eluate | 120.65 (±14.22) | 20.74 (±4.85) |

*3.2. Dynamics of Rhizosphere Microbial Communities*

The two datasets, tomato and lettuce, comprised a total of 144,000 high-quality reads for bacterial amplicons and 120,000 for fungal amplicons, with the related average coverage of 87% and 96%, respectively. These results indicate that, for both plants, a discrete part of the microbial community was captured by the sequencing.

The biodiversity was assessed through the estimation of the Simpson's D α-diversity index, whose algorithm evaluated the biodiversity based on the richness and diversity level of OTUs measured in each entry. The analysis outputs (shown in Figure S2) reported a decrease in the bacterial diversity for both lettuce and tomato, and a slight increase for the fungal diversity in tomato only.

The CCA model was applied to evaluate how the variables (N level, eluate, and the combination of the two) affected the microbial composition (Figure 3, Tomato: a, b, c—Lettuce: d, e, f). Analysis of the results shows a clear clustering according to all variables for both plants, meaning that the microbial communities were generally influenced by N level, eluate application, and their combinations. However, the shifts in the microbial composition were statistically significant only for the eluate (Figure 3b, 16.0% variance, $p = 0.001$) and the combination N70% + N30% (Figure 3c, 25.2% variance, $p = 0.003$) in

the tomato rhizosphere community. The variance of the latter includes the two variables nutrition level and the eluate, which had respective *p* values of 0.088 and 0.002, confirming that significant levels were reached only in presence of the eluate. In the case of the lettuce plant, it should be noted that a strong trend of clustering for the eluate was also observed, but the effect was not significant (Figure 3e, 11.4% variance, *p* = 0.087). This trend in lettuce plants was also found when assessing the impact on the combination of N level and eluate together (Figure 3f, 19.5% variance, eluate contribution *p* = 0.081).

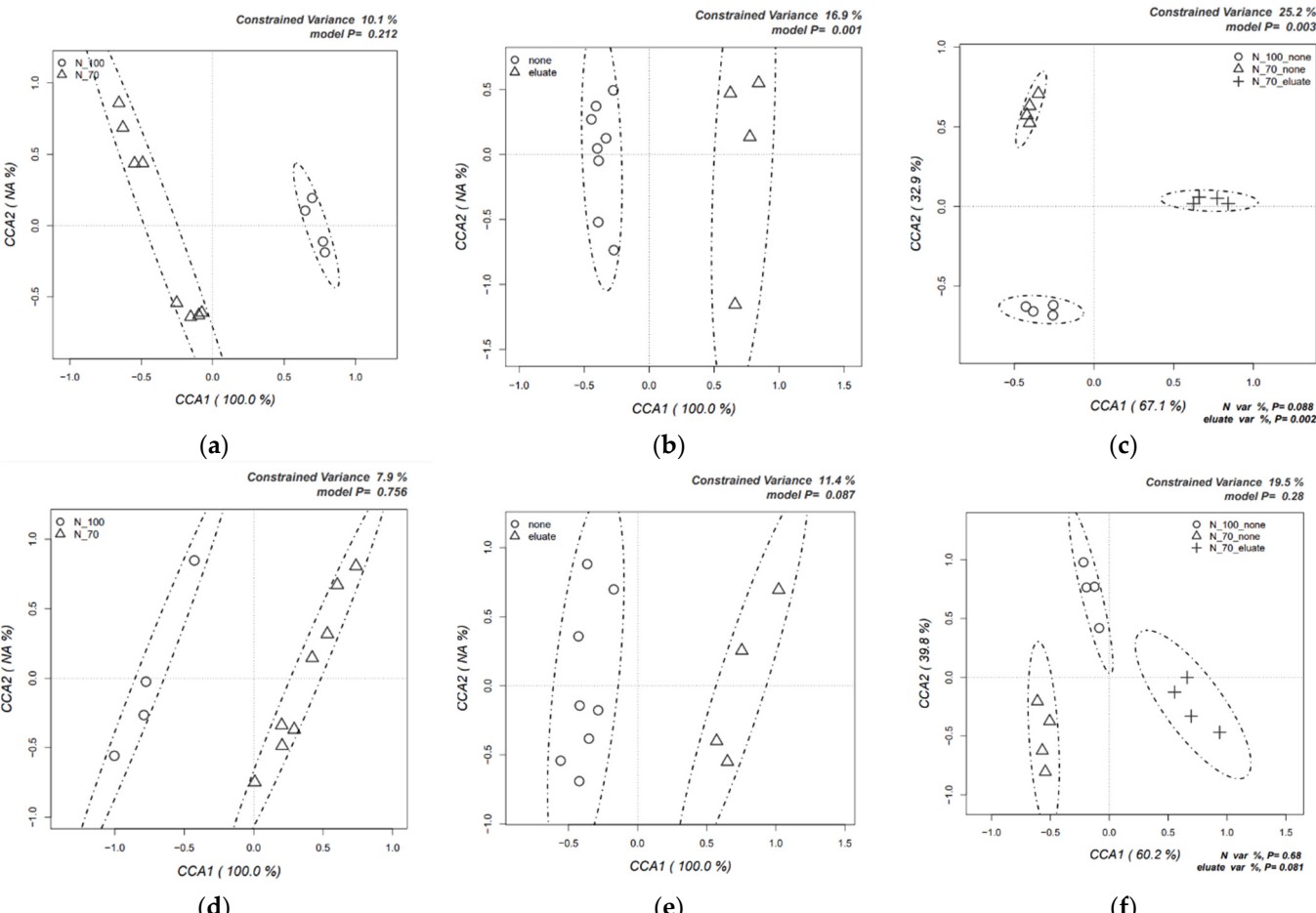

**Figure 3.** Canonical correspondence analyses (CCAs) on the impact of the eluate treatments at two levels of Nitrogen availability on the structure of bacterial communities. Dedicated *p* values are indicated on the right upper and left lower corners of the results (Upper half (**a**–**c**) is tomato. Lower half (**d**–**f**) is lettuce).

The OTUs hierarchical clustering of soil microbial communities of tomato and lettuce rhizosphere was generated at the Genus level, and the output result for bacteria is shown in Figure 4. Figure 4a clearly indicates that the eluate application on tomato caused significant changes in the rhizosphere bacterial OTUs, since N70% + N30% samples cluster separately, while N100% and N70% do not show any clustering. The clustering in the case of eluate was driven by a reduction in the *Bacillus* genus, of which a larger number of OTUs were depleted in the tomato rhizosphere samples. On the other hand, a majority of OTUs belonging to the genus *Kaistobacter* were present in the eluate samples. Furthermore, a relatively higher abundance of unclassified OTUs also contributed to the clustering of eluate samples.

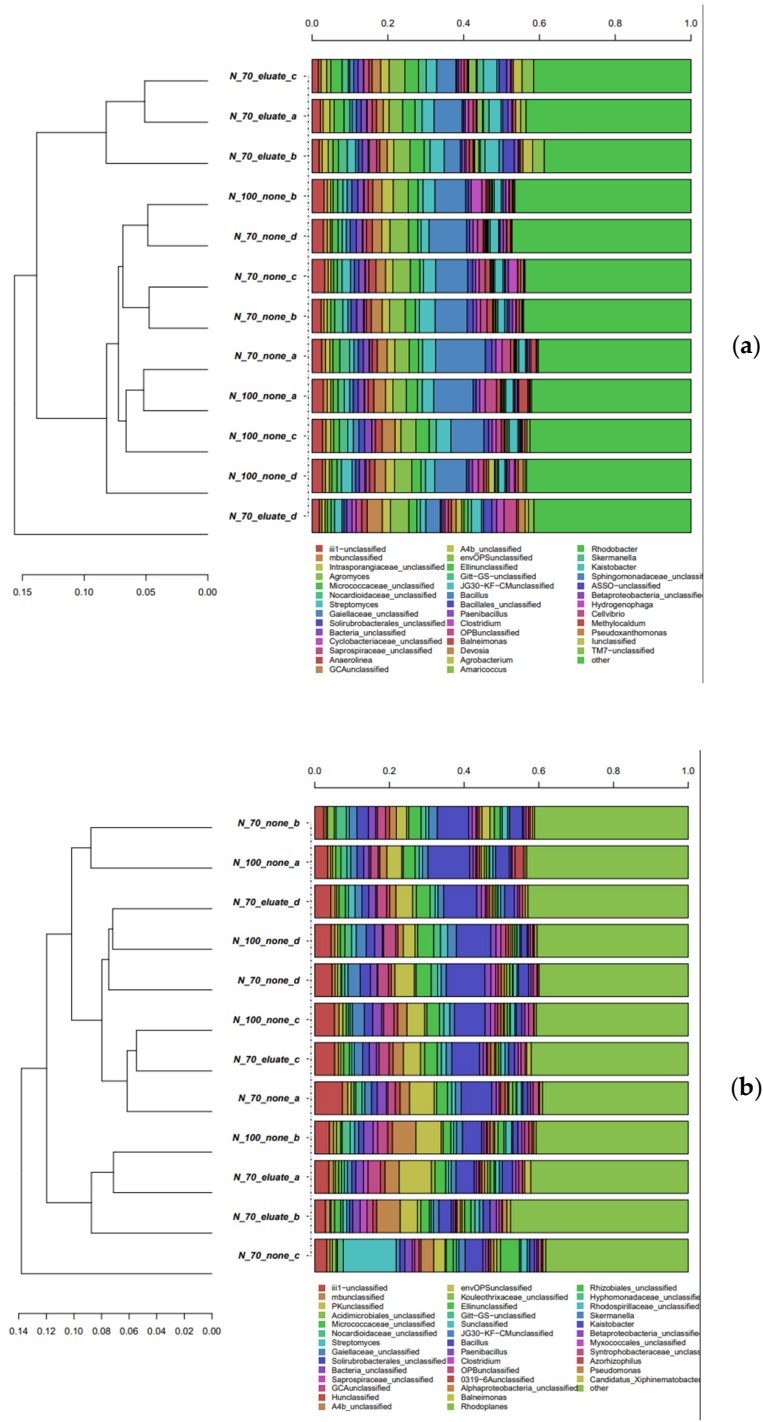

**Figure 4.** Hierarchical cluster of classified sequences using the average linkage algorithm at family classification level for bacterial taxa contributing at least 5% to a single sample for tomato (**a**) and lettuce (**b**). Taxa below the 5% threshold are attributed to the sequence group named "other".

The hierarchical clustering analysis carried out on lettuce samples did not indicate any clear clusters for the bacterial OTUs. However, a reduction in the *Bacillus* genus was also found for lettuce samples (Figure 4b). The hierarchical clustering analysis carried out on fungal OTUs of both plants did not result in clear clusters (Figure S3); this was probably due to a very high abundance of unclassified fungi, which comprised more than 50% of OTUs in some samples.

Exclusively for tomato samples, in which the most significant hierarchical clustering was observed, a detailed analysis on differently distributed OTUs was carried on with the Metastats model. This analysis enabled the identification of OTUs for which abundances were significantly affected by the treatments. As a result, a total of 11 bacterial OTUs and 21 fungal OTUs were individuated and submitted to databases for identification at the genus or species level to assess their possible ecological role in plant rhizosphere. Classification of 13 differently abundant OTUs was not possible since they belong to uncultured bacteria or unclassified fungi.

In Figure 5a are reported the differently abundant bacterial OTUs and their respective genera assigned. OTU1 showed a higher abundance, although not significant, in the presence of the eluate and was attributed to *Arthrobacter globiformis*. OTU2, attributed to *Sphingomonas sediminicola*, was significantly higher in the eluate samples. OTU3 and OTU4, belonging respectively to *Bacillus megaterium* (former *Priestia megaterium*) and *Bacillus asahii*, showed a significant reduction in the presence of the eluate. OTU7, assigned to *Knoellia subterranean*, was significantly higher in eluate samples. For the fungal OTUs (Figure 5b), OTU20098 was significantly higher in the N70% and was attributed to *Mortierella rishikesha*, and OTU77, belonging to *Nigrospora vesicularifera*, was found exclusively in N70% samples but not significantly so. Then, OTU33, assigned to *Funneliformis mosseae*, showed an increasing but not significant trend in the presence of the eluate. Finally, OTU4, assigned to *Olpidium brassicae*, was more abundant with the eluate but not significantly so.

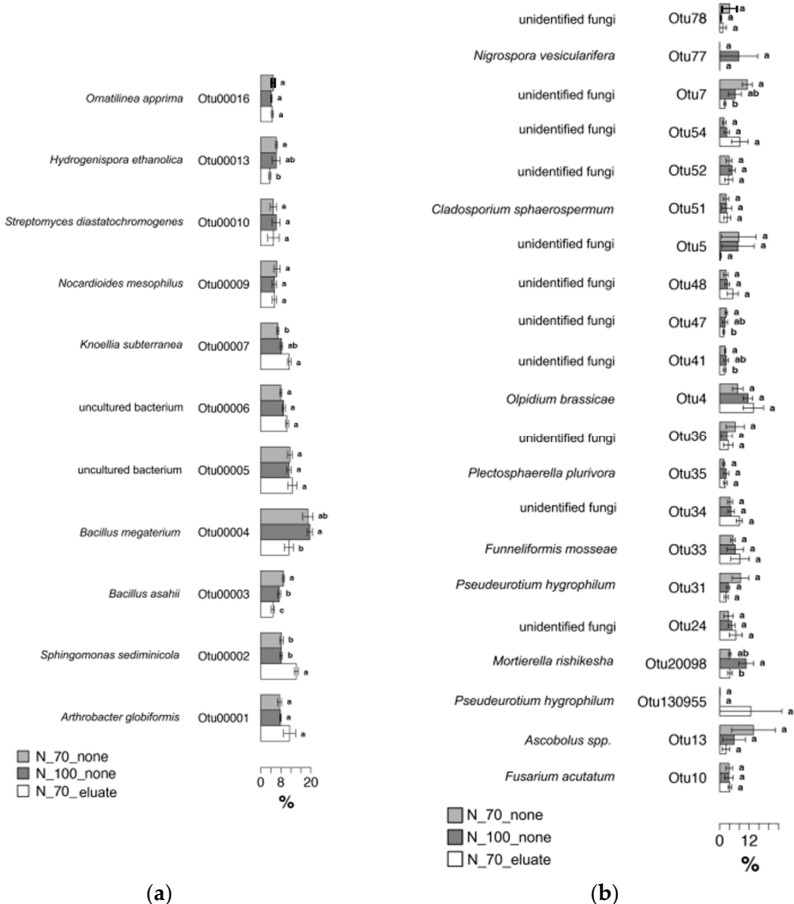

(**a**)  (**b**)

**Figure 5.** Metastats model output indicating relative abundances of the 11 most abundant bacterial OTUs (**a**) and 21 fungal OTUs (**b**) observed in tomato rhizosphere, comprising 95% of the bacterial diversity found in each treatment. OTUs showing significant differences according to statistical analysis are highlighted with different letters. The assigned genera or species are indicated for each OTU.

## 4. Discussion

The present work explores the effects of LABs eluate as an integration in mineral N fertilization on tomato and lettuce plants grown in greenhouse conditions. The eluate was considered to replace 30% of the total N, and it was applied via fertigation two times per week for a total of 13 and 27 applications for lettuce and tomato, respectively.

The trial on lettuce revealed that the eluate N can be efficiently used to reduce the input of mineral N in the cultivation of these crops. In fact, its applications divided during the entire duration of the test caused higher and faster plant growth that consequently translated into a higher final yield. Moreover, the composition of the byproduct demonstrated an additional effect on plant growth, since N70% + N30% with eluate grew more than N100%. The slightly reduced root growth in the presence of the eluate suggests that this product put plants in a more comfortable condition compared to N100%, since the reduction in provided N induced a higher root growth. In fact, Tsouvaltzis et al. (2020) [27] reported that root growth is significantly affected by N, as its reduction by 50% resulted in increased (2.5 times) root length when compared to the control.

Similar results were observed for tomato, where the lower available N produced an earlier fruit maturation, caused by an earlier flowering due to lower nutrition. As a matter of fact, it is known that applying nitrogen fertilizer delays flowering in crop species [28]. It was further observed that the eluate caused a later maturation, suggesting that plants were in a more comfortable condition than N100%. In fact, plants supplied with the eluate produced a higher number of ripe tomatoes later than all the other conditions. Moreover, these latter plants gave the highest quantity of unripe fruits, more than the comfort condition (N100%). This suggests a biostimulant effect of the eluate, the use of which allowed not only a reduction in mineral N but also improved plant performances. In fact, the N70% + N30% produced a higher plant shoot biomass compared to N100%, indicating that the eluate nourishes the plants more than full mineral nutrition. The eluate reduced root growth, but this response to N levels is species-specific [29]. In fact, the trial on lettuce gave the opposite results, with N70% producing more roots than N100%, both with and without the eluate.

The microbiological analyses revealed interesting features regarding the rhizosphere microbial composition. CCA analysis showed a significant impact of the eluate on the microbial communities of tomato rhizosphere, regardless of N fertilization levels. Similar clustering was also found in lettuce, but the differences among the treatments were insignificant. Reduction in N fertilization by 30% had an impact on clustering; however, in N70% treatment, the impact of the eluate was highlighted with a clearly visible, distant, separate cluster of N70% + N30% treatment. Together with significant changes in the hierarchical clustering of the bacterial community in tomato samples, these findings suggest that, in reduced N conditions, the eluate was effective in shaping the microbial community. This can be attributed directly to the eluate since no significant clustering between N100% and N70% was found. Results regarding the microbial composition shift suggests that the presence of the eluate reduced the total abundance of certain bacterial OTUs, such as the ones belonging to the genus *Bacillus*. This can represent a downside since the *Bacillus* genus is one of the most important genera for plant growth promotion [30,31]. On the other hand, the eluate enabled/facilitated other genera of bacteria, such as *Kaistobacter*, often detected in soil environments and often associated with biocontrol and bioremediation activities [32]. Moreover, *Kaisotbacter* is also known for its capacity to carry photosynthesis and sustain plant growth in stress conditions [33]. The shift in the microbial composition could be explained by the fact that the nutrients carried by the eluate facilitated the development of certain fast-growing microorganisms that thrive in a nutrient-rich environment [34]. Those few bacterial species able to utilize this nutrient surplus developed and prevailed, while reducing the overall bacterial diversity (as shown in Figure S2). This can be attributed to the fact that, in reduced N conditions, the eluate affected the microbial community, as also evidenced by further Metastats analysis.

In lettuce, the insignificant impact of the treatments suggests that neither the N reduction nor the eluate use influenced the microbial composition. This can be explained by the fact that 30% reduction in the mineral N is not enough to significantly affect the microbial composition of lettuce rhizosphere. A rather similar result was obtained by Li et al. (2016), in which a 20% N reduction did not lead to any changes in the microbial communities of lettuce plants. These authors suggest that the threshold for such changes was about 50%, and seasonality was another issue affecting the overall composition of bacterial communities in greenhouse conditions [35]. Moreover, the short crop cycle and the consequently shorter time for rhizosphere microbial associations with plants could be responsible for the absence of significance.

Among the most abundant OTUs in the eluate-treated tomato samples, some were assigned to beneficial microorganisms. Among the bacterial OTUs was found *Arthrobacter globiformis*, a common rhizobacterium known to possess various plant-growth-promoting properties, such as the ability to fix atmospheric N [36]. This OTU, although not statistically significant, seems to increase when the eluate is applied. The same result was also obtained in our previous study, where the presence of *A. globiformis* in bare soil was significantly increased by the eluate application [16]. Similarly, we registered a significant increase in the abundance for *Sphingomonas sediminicola*. This soil-borne bacterium is reported to induce changes in the root system structure of *Arabidopsis thaliana* and *Pisum sativum* through the production of auxins and other phytohormones [37]. In other studies, the *Sphingomonas* genus is reported to produce siderophores and to fix nitrogen [38].

Additionally, *Knoellia subterranea*, a soil-borne bacterium reported to be able to reduce nitrate to nitrite [39], was also found to be enriched in the rhizospheric soil treated with the eluate. The genus was found to be involved in the maintenance of enzyme cycles under stress conditions, therefore providing resilience to the microbial composition of plant roots under stress [40]. Contrary to our previous study on bare soil, which demonstrated a good affinity between the eluate and many Bacillus species, in this study, we found a reduction in the abundance of *Bacillus megaterium* (now *Priestia megaterium*) and *Bacillus asahii*. The former is well-studied and known for its plant-growth-promoting activities [41], while the latter is reported to support soil fertility via acceleration carbon and phosphorus cycling [42]. Collectively, beyond underlining the key role the plants have in shaping and, such as in this case, restructuring the microbial community within the bare soil, these results point to the growth promotion induced by the eluate on PGPBs, both with and without plants, and that the eluate itself has a biostimulatory effect on plants as well.

Despite a weaker effect displayed by the eluate application on the rhizosphere fungal communities, a small number of fungal OTUs were enriched in the rhizosphere. Among them was *Mortierella rishikesha,* a fungus that was previously found in rhizosphere of wheat [43]. The *Mortierella* genera is known to carry some PGP properties [44], but no data are found in the literature for *M. rishikesha* species. The eluate increased the presence of *Funneliformis mosseae*, one of the most prevalent AM fungi able to establish a symbiotic relationship with tomato plants [45]. AMF–plant symbiosis is well-known for supporting plant growth in stress conditions [46,47] and is able to shape root architecture [48,49].

Collectively, these results highlight the beneficial role of the LABs eluate and its potential application as a biofertilizer, acting both on plants and on the soil/rhizosphere interface. Further works will elucidate the mechanistic bases for the PGPM selection promoted by this solution. We speculate that the LABs eluate's metabolites, or their spent media byproducts, may selectively promote cross-feed niches [50] towards PGPMs and potentially supply plant nutrients directly [51] and indirectly (through the microbial community). Thus, it will be important to move our focus to field applications and to deepen our understanding of cross-talks between plants, microbes, soils, and biofertilizers.

## 5. Conclusions

The effect promoted by the LABs eluate application was interesting, not only for its effect on yield/biomass, but also for the trend in modulating the shoot/root growth ratio,

which could be explained with a more efficient use of the nutrient by the plant and could promote valuable crop traits if confirmed in field trials. Moreover, the eluate correlation with delayed tomato fruit ripening was also interesting, as it may affect the plant's life cycle and could possibly increase productivity. However, these trends will be confirmed by field trials and comparisons in more horticultural crops.

Additionally, HTS revealed a correlation between the eluate application and the rhizosphere microbial shift in tomato, whereas its impact was weaker in lettuce, where only some clustering trends were found. Overall, bacterial communities were more subjected to changes compared to the fugal communities in both plants. Deeper analysis of tomato rhizosphere revealed a higher presence of bacterial and fungal OTUs of importance for agriculture (although not always statistically significant), such as those of the *Kaistobacter* genus, *Arthrobacter globiformis*, *Sphingomonas sediminicola*, *Knoellia subterranean*, and *Funneliformis mosseae*.

These findings suggest the use of LABs eluate in greenhouses to integrate with mineral N fertilizer in order to stimulate plant growth and to enrich PGPM in the selected horticultural crops. Given its nature, the eluate is a sustainable ingredient that could reduce the application of costly mineral nitrogen employed in conventional agriculture, which is less available in quantity and affordability.

Further studies will aim to understand the underlying genetic and molecular mechanisms regulating both the selection on PGPMs and the resultant biostimulatory effect on crops, also validating the present results in field trials. These studies will be key not only to deepening our knowledge of microbial ecology within such agro-ecosystems (plant–microbe–soil–biostimulant), but also to building a new experimental framework and pipeline to test new circular economy products within modern agriculture.

**6. Patents**

The use of the eluate represented in this study has been protected by the patent nr. WO 2022/096723.

**Supplementary Materials:** The following supporting information can be downloaded at: https://www.mdpi.com/article/10.3390/land11091544/s1. Table S1: Effect of the eluate on plant growth. The bidimensional growth of lettuce plants was assessed. Table S2: Effect of the eluate on plant growth. Average growth values ($\pm$ s.d.) between T0–T1, T1–T2, and T0–T2 indicate that the use of the eluate to substitute 30% N is efficient in restoring and ameliorating plant growth. T0, 7 days after planting (DAP); T1, 29 DAP; T2, 50 DAP. Values $\pm$ s.d. are expressed in sqcm. Different letters differ significantly ($p < 0.05$, Duncan test). Table S3: Ripe and unripe tomato fruit number and weight. Ripe fruits were harvested 2 times, and unripe tomatoes were harvested at the end of the trial. Values $\pm$ s.d. are expressed in g. Different letters differ significantly ($p < 0.05$, Duncan test).

**Author Contributions:** G.B., conceptualization, methodology, formal analysis, data analysis, data interpretation, and writing original draft. E.T., methodology, data analysis, data interpretation, and writing original draft. S.S., methodology, experiment realization, formal analysis, and writing original draft. C.S., methodology and review and editing the draft. R.B., methodology. F.B., methodology. M.C.G., methodology. P.S.C., conceptualization, review and editing the draft, and supervision. F.V., methodology, conceptualization, validation, and review and editing the draft. E.P., conceptualization, software, formal analysis, resources, validation, writing, review and editing the draft, supervision, and project administration. All authors have read and agreed to the published version of the manuscript.

**Funding:** This research did not receive any specific grant from funding agencies in the public, commercial, or not-for-profit sectors.

**Data Availability Statement:** Not applicable.

**Conflicts of Interest:** The authors declare no conflict of interest.

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
