# Peer review of "LABs Fermentation Side-Product Positively Influences Rhizosphere and Plant Growth in Greenhouse Lettuce and Tomatoes"

_land, doi:10.3390/land11091544_

Round 1
Reviewer 1 Report
The manuscript “The application of a side stream product as fertilizer improves plant performances and shapes the rhizosphere microbial composition of lettuce and tomato in greenhouse” is well written, with a topic of interest and with an interesting application methodology. While the results obtained and consequently the conclusions appear to be less scientifically solid. Main revisions are recommended following the below indications.
Page 2, line 94: delete a "the same".
Page 3, line 123: 15 °C instead of 15 °c.
Page 4, line 129: provide more details of the fertigation system adopted.
Figures 1 and 2; make the axes and the axes title evident.
Tables 2 and 3 should report at least the dry matter content of the shoot and root or rather the dry biomass. The fresh matter values are not indicative and of little importance, while the dry matter production, as well as from a production potential point of view, is an important index of resistance to abiotic stress because it gives greater rusticity to the plant.
For the reasons mentioned above, the parts of the discussion (page 13, lines 425-434; 439-450) and conclusions (page 14, lines 524-528 and page 15, lines 537-539), referring to the results of Tables 2 and 3 and Figure 2 must be completely revised, considering the dry matter production instead of fresh matter.
Analyzes carried out on the dynamics of rirhizosphere microbial communities and microbial composition (CCA model, Hierarchical cluster and Metastats analysis) in lettuce and partly also in tomato, do not indicate a clear statistical significance. The cause of this result, in my opinion, is the limited contact time between eluate and soil to allow an alteration of the composition of the microbial rhizosphere, especially in lettuce. For this reason, my suggestion is to speak about trends and not about actual changes of microbial community composition, as reported in conclusions at page 15, lines 531-532.
Author Response
We thank the reviewer for the comments and useful insights. We have carefully revised the form, fluency and flaws highlighted in the texts. We also attach the point-per-point response to each reviews.
Best wishes,
Francesco Vuolo

Reviewer 2 Report
1. The present manuscript is an interesting piece of reading and a timely contribution. the authors have collected some unique data and provided fresh insight. I liked reading the manuscript. The authors have provided well documented study on how eluate obtained from Lactic Acid Bacteria production can be amended for lowering the N fertilizer use in agriculture. Please address the following comments for improving the quality of the manuscript-
2. Kindly provide additional insight on how the amendment of eluate obtained from Lactic Acid Bacteria production can be helpful in achieving the sustainable development goals and supporting the circular economy.
3. Please provide appropriate citations for the statements made in line 78 to line 86.
4. The term 'Nutrient Deficiency Soil' and 'Fully Fertilized Soil' should be represented as Nutrient Deficient Soil and Fertile soil, respectively.
​7​. Author should be discussing more about key findings with respect to future​ ​perspectives.
Author Response

(The authors gave the same response as above.)

Reviewer 3 Report
The manuscript titled “ The application of a side stream product as fertilizer improves plant performances and shapes the rhizosphere microbial composition of lettuce and tomato in greenhouse“. I find the idea interesting and in line with the aim of the journal. I have some concerns about the experimental setup to justify what the authors claim. Moreover, the rationale behind some of the data presented was not entirely clear. I also recommend to the authors improve their references by conducting a more extensive review of international literature. Particularly, the introduction statements are not supported by the references selected by the authors. The logic of some sentences is also questionable. Below is my point-to-point analysis of the manuscript.
The abstract is not properly written, it should be crisp, it should contain an introduction aim hypothesis aim result and conclusion. The introduction section is too long in the abstract; one line of the background of the study in the abstract attracts the reader the most. A connective link is missing between different sections. Also, the concluding part of the introduction is missing at the end of the introduction. The author should make the introduction section crisp and to the point related to research, which I don't find in the present form of the manuscript.
My main concern with a manuscript is the statical test.
what is the value of n while calculating ANOVA? n Value (3) used in the manuscript is too few to examine the normal distribution of variables in the sample, however, the Shapiro-Wilk test is appropriate for samples from 3 to 5000 but for the lesser value of n it receives the non-normal distribution. Thus ANOVA which is a parametrical test is incorrect for such small samples.
Error Bar
Secondly, the error bar in the Figures 2 AND 3 are not properly explained it seems they correspond to standard deviation which does not make any sense, that is merely for decorative purposes. I highly recommend using a 95% confidence interval instead of a Standard deviation, in the error bar.
CCA Graph is not properly explained. For example, characteristics correlated to different attribute is not explained; the Length of different vector (Variability of Variable) is not explained, etc.
Data Distribution
The author should mention the data set that passes the normality test. The author has applied parametric tests, he has not explained the reason for the parametric tests
Although the study is interesting and could be useful for a certain group of scientific fraternity, therefore, I would suggest improving the manuscript substantially, giving it a chance for the next round, because the subject is interesting. However, even an interesting subject depends on statics.
Author Response

(The authors gave the same response as above.)

Round 2
Reviewer 1 Report
lines 515-533. The authors report excessively definitive and optimistic conclusions often not fully supported by the results obtained. Reviewing these conclusions speaking more of trends than of certainties.
Author Response
We thank the reviewer for the valuable comments. We have reviewed the conclusion accodingly to what suggested. Best regards.
Reviewer 2 Report
The authors have satisfactorily addressed all the comments and the manuscript can be addressed in its present form.
Author Response
We thank the reviewer for the valuable comments.
Reviewer 3 Report
I RECOMMEND ACCEPTING MS IN PRESENT FORM
Author Response

(The authors gave the same response as above.)
